# In Vitro Crude Protein Digestibility of Insects: A Review

**DOI:** 10.3390/insects13080682

**Published:** 2022-07-28

**Authors:** María Rodríguez-Rodríguez, Fernando G. Barroso, Dmitri Fabrikov, María José Sánchez-Muros

**Affiliations:** 1Department of Applied Biology, CECOUAL, University of Almería, 04120 Almería, Spain; fbarroso@ual.es; 2Department of Applied Biology, CEImar, University of Almería, 04120 Almería, Spain; 3Department of Applied Biology, University of Almería, 04120 Almería, Spain; df091@ual.es; 4Department of Applied Biology, CEI3, University of Almería, 04120 Almería, Spain; mjmuros@ual.es

**Keywords:** hydrolysis degree, nitrogen balance, insect processing, insect meal

## Abstract

**Highlights:**

**Simple Summary:**

In order to consider insects as an alternative protein food, it is important to study the effect of digestion on their protein. The aim of this work is to collect data on the digestibility of insects and the pre-processing used to try to improve their digestibility until 2021. Limitations were found in the discussion of the data due to the diversity of methodologies used to carry out in vitro protein hydrolysis. In addition, articles evaluating the effect of insect pre-processing are very limited. Standardisation of protocols would be necessary to facilitate comparisons in future research.

**Abstract:**

The high protein content of insects has been widely studied. They can be a good food alternative, and therefore it is important to study the effect of digestion on their protein. This review examines the different in vitro protein digestibility methodologies used in the study of different edible insects in articles published up to 2021. The most important variables to be taken into account in in vitro hydrolysis are the following: phases (oral, gastric and intestinal), enzymes, incubation time and temperature, method of quantification of protein hydrolysis and sample preprocessing. Insects have high digestibility data, which can increase or decrease depending on the processing of the insect prior to digestion, so it is important to investigate which processing methods improve digestibility. The most commonly used methods are gut extraction, different methods of slaughtering (freezing or blanching), obtaining protein isolates, defatting, thermal processing (drying or cooking) and extrusion. Some limitations have been encountered in discussing the results due to the diversity of methodologies used for digestion and digestibility calculation. In addition, articles evaluating the effect of insect processing are very limited. It is concluded that there is a need for the standardisation of in vitro hydrolysis protocols and their quantification to facilitate comparisons in future research.

## 1. Introduction

In 2050, it is estimated that the world population will be more than 9000 million people. In a growing world in which the population is constantly increasing, food is a serious problem due to the high demand for food and the shortage of resources to attend to these needs [1,2]. To these needs must be added to the growing demand for proteins in farmed animals. Due to this nutritional value and ease of breeding, insects have been proposed to be protein alternatives as well as food and feed ingredients [3,4] and can be a great ally to solving world food scarcity according to the United Nations [5]. In addition, insects have a more sustainable production than traditional livestock because their feed conversion efficiency is higher, they need less water for their production, they emit fewer greenhouse gases, they prevent the deforestation of natural areas used as pasture and they are exceptionally adapted to byproduct feeding [6,7]. All of this contributes to growing interest in the study of insects as food in different fields. The European Union, through a rule set out in Annex II of Regulation 2017-893 of 24 May 2017, has made it possible to use insects for breeding farm and aquaculture animals [8]. The insect species that have been used are *Hermetia illucens* (black soldier fly), *Musca domestica* (housefly), *Tenebrio molitor* (mealworm), *Alphitobius diaperinus* (bed beetle), *Acheta domesticus* (house cricket), *Gryllodes sigillatus* (striped cricket) and *Gryllus assimilis* (two-coloured cricket). In 2021 it was also approved for use in farm animals (poultry and pigs) through Commission Regulation (EU) 2021/1372 of 17 August 2021 [9]. In addition, in November 2021 (EU Regulation 2021/1925) *Bombyx mori* was added as the eighth species approved [10]. The European Union has made further progress in the use of insects as a novel food by first allowing human consumption commercialisation of *Tenebrio molitor* larvae (Implementing Regulation EU 2021/882) [11] and later in frozen, dried or powdered forms of three species: *Locusta migratoria, Tenebrio molitor* and *Acheta domestica* (Implementing Regulations EU 2021/1975, 2022/169 and 2022/188, respectively) [12,13,14].

In general, insects have a high protein proportion; nevertheless, there is great variability in their protein content (Table 1). For example, TM crude protein ranges from 46 to 69%, and HI varies between 26 and 61%. The origin of this huge variability could be due to their stage of life [6,15,16,17], their diet or ecology [18], or their sex because female crickets have more fat and less protein than males [19]. In addition to these various factors, the way in which they are processed (thermal and mechanical) should be added.

As a protein source, in addition to the crude protein content of insect meals, it is necessary to know the balance of essential and nonessential amino acids and their bioavailability. The amino acid profile of insect meals has been extensively studied and shows great variability depending on the order and the species to which they belong [29] or the treatment the insect has received [3,39]. Barroso et al. [29] found a relationship between order and amino acid profile, in addition to the similarity of order dipterans to fishmeal with high levels of phenylalanine, tyrosine and valine, histidine, lysine, threonine and methionine, although they were deficient in leucine. In contrast, Orthoptera and Coleoptera were not deficient in leucine, although their profile was very different from that of fishmeal, with lower levels of histidine, lysine and threonine. On the other hand, Huang et al. [3] studying the amino acid profile of HI after cooking in conventional and microwave ovens, found differences in its amino acid profile, with aspartic acid (11.01/100 g protein) predominating in conventional ovens compared to glutamic acid (12.65/100 g protein) in microwave ovens. Along the same lines, Janssen et al. [39] studying the profile of coleoptera (TM and *Alphitobius diaperinus*), found a decrease in hydrophobic amino acids (isoleucine, leucine, tyrosine, phenylalanine and valine) after blanching and an increase in nonhydrophobic amino acids (cysteine, glutamic acid and glutamine) compared to unroasted control samples.

Bioavailability depends on the digestibility of the protein source, and this depends largely on the degree of protein hydrolysis after digestion and the amino acid absorption intestine capacity.

Determining protein source in in vivo digestibility is a long process that involves the rearing of animals on a specific diet, faecal collection and its analysis. It is very costly when the digestibility of various sources is to be determined and involves the use of many animals. The development of in vitro protein hydrolysis techniques allows a protein source digestibility estimation without the disadvantages of in vivo digestibility [40]. Likewise, in vitro digestibility techniques are used to obtain protein hydrolysates that allow greater bioavailability of amino acids.

Digestion and absorption processes in animals are very complex and dynamic; therefore, simulating a complex process using in vitro methods is difficult, and the results will never be as accurate as in vivo methods, but is a great alternative, taking into account the complexity and time needed for the experiments compared to in vivo methods.

In a recent study evaluating the inclusion of defatted black soldier larvae meal in extruded dog food, Penazzi et al. [41] studied whether the in vitro digestibility method could be comparable to the in vivo method. They found that although the estimates were higher than those obtained by in vivo analysis, the in vitro digestibility values for crude protein (82.3%) were similar to the in vivo results (80.1%) and considered that this method could be a possible alternative to the traditional method of total faecal collection in in vivo digestibility tests.

In vitro protein hydrolysis methods mimic the conditions simulated by digestive processes through proteolytic enzymes by measuring the percentage of proteins that are hydrolysed by these enzymes [42]. These methods are quickly reproducible, give a great digestibility estimation for a wide range of foods [43] and allow comparison between ingredients when used under the same experimental conditions.

Therefore, the aim of the present review is to study the different methods and applications of in vitro protein hydrolysis of insects performed thus far and to assess the different processing techniques of meals to aid in insect protein hydrolysis to provide an overview of the knowledge obtained thus far.

## 2. Study Summary

Two databases were used: Science Direct and Scopus. The following keywords were entered: “Protein hydrolysis”, “Digestibility”, “In vitro hydrolysis”. Inclusion criteria were insect-based articles with in vitro protein hydrolysis. Exclusion criteria were articles on in vivo hydrolysis, bioactive peptides (pharmacology/health field) and whose results were qualitative.

Finally, 34 articles (30 experimental and 4 reviews) published up to 2021 related to in vitro protein hydrolysis of protein from different species of insects were studied. These articles differ according to the type of insect, the way to obtain the insect, the hydrolysis method used, the hydrolysis phases and the enzymes involved in each phase. Differences are also found in the way to verify protein hydrolysis after hydrolysis both in the methodology used and measurement units. In this review, the results were divided into two types of hydrolysis. In vitro protein hydrolysis as a simulator of digestive processes (oral, stomach and intestine) in the different insects used in feed and the effect of different meal processing of insect meals to improve/evaluate the digestibility of insects.

## 3. Methods Used for Digestion and for the Determination of In Vitro Digestibility

In vitro digestibility is a laboratory-based methodology that attempts to simulate digestive processes that occur in humans or animals. It is a less expensive and slow alternative to in vivo digestibility, which analyses the differences in composition between ingested food and faeces after the digestion process. However, although in vitro digestion attempts to simulate in vivo digestive processes, these methods can hardly fully mimic the actual pH and temperature conditions of the digestive system [44]. For this reason, digestibilities obtained in vitro usually have lower values than in vivo digestibility realised with animals [43].

Protein digestion begins in the stomach with pepsin action continuing in the intestine with the digestion of trypsin and chymotrypsin and is completed by the action of proteases on the intestinal surface [45]. Despite this, due to the complexity of the in vitro digestion simulation, some authors include, in addition to the gastric and intestinal phase, an oral phase (with amylases or artificial saliva). Depending on the phases included in the in vitro stimulation, the results can vary significantly. For example, Yi et al. [30] found that the protein digestibility of TM was 54% when both phases were performed (with pepsin and pancreatin) and only 38% when only the gastric phase was performed (with pepsin).

As seen in Table 2, with insect meals, most researchers use both phases (gastric and intestinal). However, some researchers have carried out simple gastric digestion [46,47,48,49,50,51,52], and Séré et al. [53] only performed enteric digestion. Only a few have previously added an oral phase [2,3,54,55,56].

The enzymes used are usually of commercial origin. There are no major variations between studies, mainly using α-amylase in the oral phase, pepsin in the gastric phase, and pancreatin in the intestinal phase. There are minor modifications; for example, Azzollini et al. [54] did not use α-amylase in the oral phase because following Woolnough et al. [67] they considered that salivary hydrolysis of starch was negligible compared to that degraded by pancreatin in the intestinal phase. In some work, duodenal digestion of pancreatin has also been combined with trypsin or bile extract.

Other variables are important, such as pH, temperature and time. For the oral phase, the pH remains virtually neutral; in the gastric phase, it varies in an acid range between 2–3, and in the intestinal phase, a neutral-alkaline pH of approximately 7–8 is mostly established. The temperature during the process remains constant over a range that can vary between 37–39 °C. Ramos-Elorduy et al. [52] reached 45 °C. The incubation time for each phase is highly variable. As a rule, the oral phase lasts a few minutes, whereas the gastric and intestinal phases last a few hours.

There are many methods used to evaluate the digestibility percentage. In general, in vitro protein digestibility is often determined by two principal methods:

(1) Nitrogen balance: the difference between the amount of nitrogen ingested and the amount of nitrogen present in the final undigested substrate is evaluated. The formula is used:Digestibility (%) = (A − B)/A × 100%
where A is the N content in the original sample before digestion and B is the N content in the final sample after digestion.

As shown in Table 2, in the study of insect digestibility, nitrogen balance is the most commonly used method.

(2) Hydrolysis degree: values of the number of peptide bonds cleaved during hydrolysis with respect to the total number of peptide bonds and studied in the supernatant. In this case, the formula is used:
$$\mathrm{Hydrolysis}\mathrm{degree}:\;\%DH=\frac{h}{htot}*100$$
where *h* are the hydrolysed peptide bonds of the sample, and *htot* is the total peptide bonds. To obtain the number of total amino groups, complete hydrolysis was performed (6 N HCl at 110 °C for 24 h).

In turn, for the determination of the protein hydrolysis degree, there are several methods [68]. This can be quantified by determining the amino groups released during hydrolysis using compounds that react specifically with amino groups such as trinitrobenzene sulfuric acid (TBNS), o-phthaldialdehyde (OPA) or ninhydrin.

Among the researchers who determined the degree of hydrolysis of insects, it seems to be predominantly quantified by OPA, as only Zielińska et al. [2,56] have used the reaction with TBNS.

Another method to determine the degree of hydrolysis is the *pH-stat/drop method*, which is based on determining the release of protons produced when peptide bonds are cleaved by the action of enzymes [69]. This release causes a pH change (acidification) of the reaction medium, and a base must be added to maintain the pH. For the calculation of protein digestibility in the Hemiptera *Carbula marginella* and the Lepidoptera *Cirina butyrospermi*, Séré et al. [53] used the following formula:Digestibility = 4.33 + 53.21X 
where X is the volume of NaOH (mL) poured at t = 10 min to maintain the pH at 8.0. Nielsen et al. [68] indicated that the pH-stat technique should be used in pH conditions above 7, and Bryan and Classen [69] pointed out that this method is not suitable for determining food digestibility in terrestrial animals but is especially suitable for aquatic nutritional research. In their review, these authors consider that the fish tract is simpler and that highly digestible feeds, such as fishmeal, are used in aquaculture; therefore, high digestibility correlations are obtained.

Finally, Huang et al. [3] used two methodologies (nitrogen balance and degree of hydrolysis) to evaluate the effect of the drying method on the digestibility of amino acids of the black soldier fly (*Hermetia illucens* L.), without finding major differences in the results of both methods.

## 4. Insect Meal Digestibility

Table 3 summarises the results obtained on the in vitro protein hydrolysis of insects by various researchers. Large differences are observed, which can be attributed to the method of digestion, the method used to assess digestibility and the species or the developmental stage of the insect [55]. However, despite these differences, the protein digestibility of insects can be considered to be high; most species are between 80% and 90%, and even the larva of the lepidopteran *Laniifera cyclades* is reported to have 98.9% protein digestibility [51]. Few studies have compared insects with other reference feeds, but those analysed have shown that insects show a similar digestibility to fish meal (84.9% [46]; 85,7% [58]) and are a slightly lower digestibility than soybean meal (95%) [58].

In a study that used crude enzyme extracts from the digestive tracts of ducks, Kovitvadhi et al. [60] compared the in vitro protein digestibility of 17 insect species with fishmeal and soybean meal. The obtained *Z. morio*, TM, *M. domestica*, *B. mori* (pupae), and even the cockroach *Periplaneta americana* showed similar or higher digestibility than fishmeal or soybean meal. In addition, they found that acid detergent fibre (ADF) was the best predictor of protein digestibility, observing a significant negative correlation between these two parameters. This fact was observed by Marono et al. [63] who argued that the enzymes of monogastric animals are very inefficient at digesting ADF, and for better results, it is very important to obtain the chitin content.

Chitin is a polysaccharide of glucosamine and N-acetylglucosamine containing N in its molecules [6], is embedded in a scleroprotein matrix [70] and is neither degraded nor absorbed in the small intestine [71]. This has two implications: first, chitin may have “anti-nutritional” properties due to its potentially negative effects on protein digestibility [72], and second, there is no exact correlation between crude protein content and biologically available nitrogen [66], as N (used to estimate CP) is a major component of the indigestible cuticle of insects.

Specifically, Ozimek et al. [73] found that the whole dried honeybee contained up to 11% chitin, which in turn contained up to 6.9% N. Furthermore, they concluded that chitin removal improved protein digestibility, amino acid content, protein efficiency ratio and net protein utilisation.

However, although chitin can be considered indigestible, this depends on the enzyme package of the consuming species. Thus, several studies have observed that chitin/chitosan can be partially digested in humans due to the presence of chitinolytic enzymes from bacteria located in the gastrointestinal tract [74,75]. Another interesting aspect of chitin from the exoskeleton is that, as suggested by Lee et al. [76] it may have a positive effect on health, as it can stimulate the immune system.

Jayanegara et al. [77] evaluated the effect of decreasing the chitin content of cricket (*Gryllus assimilis*) through exoskeleton reduction (manual removal of head, legs and wings) or extraction with chemical solvents on digestibility. Despite the significant decrease in chitin with manual removal of the exoskeleton or its disappearance with chemical extraction, these authors did not obtain differences in the in vitro organic matter digestibility. However, it should be noted that the fermentation was carried out in rumen liquid in a first step and pepsin in a second step and that they did not determine the protein digestibility.

In addition to chitin, other processes may influence the hydrolysis of insect proteins, such as the autolysis observed by Yi et al. [30] who found a high content of initial NH_2_-free groups in insects. This may be a consequence of the action of insect digestive peptidases in their own body.

It should also be noted in Table 3 that different studies involving the same species, such as Kovitvadhi et al. [60], Marono et al. [63] (TM and HI) and Poelaert et al. [66] (TM and *A. domesticus*), have obtained lower values in the in vitro crude protein digestibility (IVCPD). However, these authors also obtained a low digestibility in all the feeds analysed, with soybeans having an IVCPD of 60.8% [65] or 61% [60], while Bosch et al. [58] observed 94.7%.

### 4.1. Order Blattodea

In general, these species seem to have the lowest digestibility (Table 3), and some species do not reach 80%. According to Bosch et al. [20] the chitin content and sclerotization of their exoskeleton may be the most decisive factor.

### 4.2. Order Diptera

The most studied diptera are HI and *Musca domestica*. Both show similar digestibility data with maximums of 89.7% [57] and 93.3% [58], respectively, in their larval stages. In contrast, HI prepupae have 50% of digestibility. This could be explained by the correlation between ADF and chitin. In addition, the proportion of ADF changes according to the life stage of the insect, being higher in the adult stage than in the larvae [60].

### 4.3. Order Coleoptera

Several studies have found that coleopteran larvae show similar or slightly higher digestibility percentages than dipterans, such as *Alphitobius diaperinus*, TM and *Zophobas morio,* with digestibility ranging from 85.0% to 92.5% [31,57,58]. This higher digestibility may be related to the chitin content. Yang et al. [66] found that the adult edible beetle *Holotrichia parallela*, which has a higher proportion of chitin (10%), shows a lower digestibility (78.3%). On the other hand, Finke [78] estimated a higher chitin content in HI (dipteran) with 5.4% chitin than in TM (coleopteran), which contains only 2.8% chitin.

### 4.4. Order Lepidoptera

Lepidoptera larvae are one of the groups with the highest digestibility, above 90%. *Cirina butyrospermi* was the only exception, with approximately 82% digestibility obtained with the pH-stat method, while the nitrogen balance method was used for the other lepidoptera species.

### 4.5. Order Orthoptera

Similar to the other groups, studies on the digestibility of Orthoptera species are scarce. *A. domesticus* has a high digestibility, with 85.3% [50] being the lowest and 91.7% [58] the highest. Ndiritu et al. [50] only performed the gastric phase. The other orthoptera analysed did not show digestibility lower than 82.3%.

## 5. Effect of Insect Processing on Protein Digestibility

The processing step can influence the quantity and quality of bioavailable protein [79]. There are various methods for obtaining and processing insect meals, either to improve the digestibility of the insect meal, to obtain bioactive peptides (BAPs) (in the pharmacological field), or as protein hydrolysates (in the food industry).

If the use and marketing of insect meal in animal feed and human food are to be promoted, it is essential to investigate which processing methods are the most appropriate; Kinyuru et al. [59] state that insect processing methods can affect the nutritional potential of insects, especially the digestibility of their proteins. Pretreatment is considered to be any processing that has been carried out on insects from slaughter to their use as a substrate for in vitro hydrolysis. Table 4 lists the articles that have carried out some modification/processing on insects prior to hydrolysis.

### 5.1. Intestinal Removal

For those species consuming fibrous plants, such as the Mophane worm (*Imbrasia belina*), one option by Madibela et al. [80] was to degut the larvae. They found that degutted larvae were significantly more digestible than nondegutted larvae. The authors suggest that undigested Mophane leaves within the gut of the nondegutted Mophane worm decreased digestibility.

### 5.2. Slaughter Method

Normally, insects are slowly killed by freezing [81], which causes enzymatic browning of insect protein fractions due to melanisation, with endogenous phenoloxidase playing a key role in this browning [82]. Phenoloxidase-induced modifications could affect not only the visual organoleptic appearance of the meals but also their protein quality, such as the loss of cysteine and lysine [49], extractability and digestibility [83]. Therefore, the protein digestibility of insects could be expected to be reduced by this browning, as in the case of plant proteins [82].

Enzymatic browning can be inhibited by chemical inhibitors, (e.g., ascorbic acid, sulphite, proteolytic enzymes) or by physical treatment, (e.g., blanching, ultrafiltration, sonication) [84]. Therefore, Leni et al. [49] evaluated how the method of slaughter (freezing or blanching) affected the digestibility of HI. They found that blanching inhibited the enzymatic browning process, decreasing the loss of amino acids and increasing the enzymatic digestibility of the larvae. According to these authors, in addition to blocking melanisation, blanching favours enzymatic digestion because it probably also denatures proteins.

In contrast, in a similar study with *Acheta domesticus*, TM and HI, Janssen et al. [82] used sulphites and blanching as a method to inhibit browning but found no differences, compared to the control (no pre-treatment), in the degree of hydrolysis in the enteric phase (trypsin), although browning did seem to affect the gastric phase (pepsin) in TM and HI.

### 5.3. Protein Isolation

Séré et al. [53] compared the protein digestibility of defatted *Carbula marginella* and *Cirina butyrospermi* meals with isolate protein from the same insects. They found that protein digestibility was significantly higher in the isolate protein and justified this by the elimination of chitin in this fraction, as chitin reduces digestibility.

### 5.4. Defatting

Lee et al. [61] investigated the protein digestibility of whole and defatted (with 70% ethanol) meal of *Protaetia brevitarsis* larvae and compared it with that of beef loin. They found that although defatting induced a decrease in α-aminogroup content, no significant differences in in vitro protein digestibility were observed between the three samples. Although in vitro protein digestibility was not improved, defatted larvae showed higher microbial safety.

### 5.5. Heat Processing

Huang et al. [3] evaluated how the drying method at 60 °C in a drying oven to constant weight vs. in a microwave oven at 500 W for 15 min affected the amino acid digestibility of HI larvae. They found that the digestibility of larvae dried in a conventional drying oven was superior to those dried in a microwave oven. They were of the opinion that the latter was more difficult to digest because the high temperature of the microwave could cause polymerisation of the protein particles.

### 5.6. Cooking Techniques

These are used to increase the safety and shelf life of insects [85] and to improve the sensory characteristics of foods [62], but as a drawback, they can lead to the production of anti-nutritional or toxic elements [86]. Several studies have subjected insects to different heat processing to try to improve their digestibility (Table 4), but the results are unclear and even contradictory. According to Opstvedt et al. [87], insect protein digestibility can be increased if insects are subjected to a denaturing temperature, which allows digestive enzymes to act on the unfolded polypeptide chain. Caparros et al. [31] evaluated different heat processing methods (vacuum-cooked, fried, boiled, and baked 15–30 min in an oven at 70.0 °C) in TM larvae, and found that all (except fried) larvae significantly improved protein digestibility compared to control (uncooked) larvae.

In contrast, Mancini et al. [62] studied the effect of cooking techniques (oven cooking 70 °C for 30 min, oven cooking 150 °C for 10 min, deep frying, pan frying, microwaving, boiling and steaming) in TM larvae, and found that all processes decreased protein digestibility in relation to control larvae. According to Opstvedt et al. [87], excessively high temperatures can reduce protein digestibility by inducing amino acid reactions that hinder enzymatic digestion. Only oven cooking at 70 °C for 30 min showed a significant increase in digestibility.

Additionally, Poelaert et al. [65] studied the effect of autoclaving and oven cooking (150 °C for 30 min and 200 °C for 10 min) on TM and *A. domesticus*, and found that heat treatments did not improve but rather negatively affected IVCPD. In all cases, the digestibility of the insects decreased with the different heat treatments. Caparros et al. [31] attributed this to the higher temperatures used by Poelaert et al. [65] (150 and 200 °C) compared to theirs (70 °C). Caparros et al. [31] considered that this causes protein oxidation and an increase in disulfide bonds, which hinders the action of enzymes [88].

Similarly, David-Birman et al. [89] studied the impact of thermal processing on the cricket *A. domesticus* meal, found that cooking had no great effect, but that roasting increased the proteolytic breakdown of cricket proteins. Manditsera et al. [55] studied cooked *Eulepida mashona* and *Henicus whellani* insects, and obtained a higher IVCPD in raw insects than in boiled and roasted insects. Madibela et al. [80] studying the Mophane worm (*Imbrasia belina*), did not obtain conclusive results on the effect of roasting with respect to the control group, since the IVCPD varied depending on whether the larvae were degutted; however, in all cases, the boiled larvae were less digestible than the control. To add further complexity to this evaluation of cooking, it appears that the effect may also depend on the insect species. Thus, Zielińska et al. [56] studied the effect of boiling (100 °C–10 min) and baking (oven at 150 °C–10 min) on TM and the orthoptera *Gryllodes sigillatus* and *Schistocerca gregaria* and obtained very different results depending on the species. While digestibility increases with boiling, with respect to raw, in TM and *S. gregaria*, it decreases in *G. sigillatus*, and the opposite happens with baking, which does not affect *S. gregaria*, and decreases digestibility in TM and increases digestibility in *G. sigillatus*. Similarly, Kinyuru et al. [59] studied how toasting or solar drying affected the digestibility of winged termite (*Macrotermes subhylanus*) and grasshopper (*Ruspolia differens*), and found that while termites were not affected by the treatments, fresh grasshoppers had higher protein digestibility than toasted and dried ones, the latter being less digestible.

All studies seem to agree that protein digestibility significantly decreases when insects are fried [31,62]. According to Caparros et al. [31] when frying, larval lipids oxidised, they complexed with proteins, and according to Hęś [90] these lipid–protein complexes are less suitable for the enzymatic action of proteases.

### 5.7. Extrusion

Extrusion is a widely used feed processing technique in animal nutrition. Its benefits include improving the nutritional value of feed by destroying anti-nutritional factors and undesirable enzymes [91,92], thereby improving feed digestibility [93] and increasing the solubility of dietary fibres [91].

Although the previous literature reported a significant loss of protein digestibility in diets containing insects [30,63], Azzollini et al. [54] when incorporating up to 20% TM in extruded snacks, found that digestibility was not affected, remaining at an average value of 90.2%. They even pointed to an improvement in the digestibility of the proteins of TM larvae added, possibly because the mechanical shear forces generated during extrusion were able to mechanically break the protein bonds of the sclerotized proteins attached to the exoskeleton. However, more recently and in TM, Cho et al. [47] evaluated the effect on protein digestibility in feeds with different degrees of inclusion of this insect (0%, 15% and 30%), extrusion temperature (140 °C and 150 °C) and humidity (40 and 50%). It seems clear that greater inclusion of TM increases digestibility. However, with regard to temperature and moisture, the results seem inconclusive. The positive or negative effects of temperature and moisture depend on the degree of TM inclusion (15% or 30%) and contradict each other.

Irungu et al. [48] also evaluated aquaculture feeds extruded at 20 or 30% moisture, where shrimp meal was substituted with different proportions (25, 50 and 75%) of *Acheta domesticus* or HI. They found inconclusive results and obtained a significant interaction between the type of insect meal, the level of substitution and the moisture content. However, they found that with extrusion, the crude fibre content decreased. According to Mogilevskaya et al. [94] the pressure and shear developed during extrusion could cause plastic deformation of the chitin, dispersing it and thus increasing the specific surface area of its particles, which facilitates the degradation of this polymer. In general, these authors found a lower digestibility with extruded feeds containing HI, which they attributed to the effect of fat. According to these authors, fat could inhibit pepsin activity.

Ottoboni et al. [64] studied different mixtures of HI larvae or prepupae and wheat flour. Furthermore, extrusion was carried out at different temperatures (60, 69, 80 and 91 °C). These authors found that the extrusion process increased the in vitro digestibility of organic matter (OMD) but not the in vitro digestibility of crude protein (DCP) compared to the nonextruded control. This increase in OMD may be because extrusion (pressure and heat) causes gelatinisation of wheat starch [95], which increases its digestibility [96]. Temperature does not seem to affect digestibility. Although it does not seem to improve protein digestibility, these authors consider that when larvae are destined for feed production, extrusion can improve the processing of these larvae (with high fat and moisture content), as it limits steps such as defatting and drying.

In summary, it does not seem to be generalisable that extrusion has a clear positive effect on the digestibility of feed containing insect meal. The studies are inconclusive and even contradict each other. The effect of extrusion depends not only on the species, the degree of inclusion, pressure, temperature and humidity but also on frequent interactions between these parameters, which makes it even more difficult to draw conclusions.

## 6. Limitations

This review has some limitations. First, in some cases, comparisons of protein hydrolysis results may be imprecise or biased. This is due to the great diversity of methodologies used for digestion and the different methods for assessing protein digestibility In vitro. As far as possible, we have tried to compare results even though they were not obtained with the same methodology. Second, in agreement with Bosch et al. [58] since protein digestibility is a fundamental aspect of nutrition, there are so few articles that study this parameter in insects, despite the large number of species consumed worldwide. Publications evaluating the effect of insect meal processing on IVCPD are also limited. More research is therefore needed to be able to make absolutely certain statements on this subject.

## 7. Conclusions

Based on the research summarised in this review, it could be concluded that the high protein digestibility of the insect meals proves the suitability of the application of these meals in the formulation of food or feed products. Although insects, in general, have a high IVCPD, a careful choice of species is necessary, as there are great differences between them. With regard to the processing of meals, further knowledge of their effects and interactions is needed, because as Poelaert et al. [65] point out, it is necessary to avoid possible detrimental consequences on digestibility. Another conclusion that emerges is the need to standardise protocols when developing methodology and obtaining results. This would allow direct comparisons to be made for all future research.

## Figures and Tables

**Table 1 insects-13-00682-t001:** Crude protein content (%DM) of the insect species most frequently used as food.

Specie	Stage	CP (%DM)	Source
*Tenebrio molitor*	Larva	46–69	[2,15,16,20,21,22,23,24,25,26,27,28,29,30,31]
*Hermetia illucens*	Larva	40–60.8	[3,32,33,34,35]
*Musca domestica*	Pupa	40.1	[29]
*Musca domestica*	Larva	46.9	[29]
*Alphitobius diaperinus*	Larva	58.0	[28]
*Bombyx mori*	Larva and Pupa	45–69.8	[28,36,37]
*Gryllodes sigillatus*	Adult	61.3–70	[2,38]
*Schistocerca gregaria*	Adult	76.0	[2]

CP: crude protein. DM: dry matter.

**Table 2 insects-13-00682-t002:** Summary of methods and enzymes used in the study of insects’ meal digestibility.

Author	Oral Digestion	Gastric Digestion	Duodenal Digestion	ICPD Method
[46]		Pepsin		N balance
[54]	Simulated salivary	Pepsin	Pancreatin + bile extract	DH (OPA)
[57]		Pepsin	Pancreatin	N balance
[58]		Pepsin	Pancreatin	N balance
[31]		Pepsin	Pancreatin	N balance
[47]		Pepsin		N balance
[3]	α-amylase	Pepsin	Pancreatin + bile extract	DH (aa)
[3]	α-amylase	Pepsin	Pancreatin + bile extract	DH (aa)
[48]		Pepsin		N balance
[39]		Pepsin	Trypsin	DH (pH-Stat)
[59]		Pepsin	Pancreatin	N balance
[60]		Enzyme extracts from gastric tracts from ducks	Enzyme extracts from duodenal tracts from ducks	N balance
[61]		Pepsin	Pancreatin + lipase + bile extract	DH (OPA)
[49]		Protease from *Bacillus licheniformis*		DH (OPA)
[62]		Pepsin	Pancreatin	N balance
[55]	Simulated salivary	Pepsin	Pancreatin + bile extract	DH (OPA)
[63]		Pepsin	Trypsin + pancreatin	N balance
[50]		Pepsin		N balance
[64]		Pepsin	Pancreatin	N balance
[41]		Pepsin	Pancreatin	N balance
[65]		Pepsin	Pancreatin	N balance
[51]		Pepsin		N balance
[52]		Pepsin		N balance
[53]			Trypsin + chymotrypsin + peptidase	DH (pH-Stat)
[66]		Pepsin	Pancreatin	N balance
[30]		Pepsin	Pancreatin	DH (OPA)
[2]	α-amylase	Pepsin	Pancreatin + bile extract	DH (TNBS)
[56]	α-amylase	Pepsin	Pancreatin + bile extract	DH (TNBS)

CPD: in vitro crude protein digestibility. DH: Degree of hydrolysis. DM: dry matter. N: nitrogen. aa: amino acids.

**Table 3 insects-13-00682-t003:** In vitro crude protein digestibility of insects.

Order	Specie/Ingredient	Stage	% Digest.	Phase	ICPD Method	Author
Blat.	*Blaberus craniifer*	Adult	78.4	G-I	N balance	[58]
	*Blaptica dubia*	Adult	83.8	G-I	N balance	[58]
	*Eublaberus distanti*	Adult	76.4	G-I	N balance	[58]
	*Periplaneta americana*	Nymph	72 *	G-I	N balance	[60]
Coleop.	*Alphitobius diaperinus*	Larvae	91.5	G-I	N balance	[58]
	*Holotrichia parallela*	Adult	78.3	G-I	N balance	[66]
	*Tenebrio molitor*	Larvae	60 *	G-I	N balance	[62]
	*Tenebrio molitor*	Larvae	66.1	G-I	N balance	[63]
	*Tenebrio molitor*	Larvae	80 *	G-I	N balance	[60]
	*Tenebrio molitor*	Larvae	92.5	G-I	N balance	[57]
	*Tenebrio molitor*	Larvae	91.3	G-I	N balance	[58]
	*Tenebrio molitor*	Larvae	85	G-I	N balance	[31]
	*Tenebrio molitor*	Larvae	72.5	G-I	N balance	[65]
	*Zophobas morio*	Larvae	92	G-I	N balance	[58]
	*Zophobas morio*	Larvae	77 *	G-I	Nbalance	[60]
Dipt.	*Hermetia illucens*	Larvae	67.3	G-I	N balance	[63]
	*Hermetia illucens*	Larvae	87.7	G-I	N balance	[57]
	*Hermetia illucens*	Larvae	89.7	G-I	N balance	[58]
	*Hermetia illucens*	Larvae	81.6	G	N balance	[46]
	*Hermetia illucens*	Prepupa	50 *	G-I	N balance	[60]
	*Hermetia illucens*	Pupae	77.7	G-I	N balance	[58]
	*Musca domestica*	Larvae	93.3	G-I	N balance	[57]
	*Musca domestica*	Larvae	84.3	G-I	N balance	[58]
	*Musca domestica*	Larvae	73 *	G-I	N balance	[60]
Hemiptera	*Atizies tascoensis*	Adult	89.3	G	Nbalance	[51]
Hyme.	*Atta mexicana*		87.6	G	N balance	[51]
	*Brachygastra mellifica*		85.2	G	N balance	[52]
	*Liometopum apiculatum*	H-L-P	93.9	G	N balance	[51]
	*Macrotermes subhylanus*	Adult	90.5	G-I	N balance	[59]
	*Polybia parvulina*		86.4	G	N balance	[52]
	*Vespula squamosa*		76.6	G	N balance	[52]
Lepid.	*Bombyx mori*	Larvae	40 *	G-I	Nbalance	[60]
	*Bombyx mori*	Pupae	72 *	G-I	Nbalance	[60]
	*Cossus redtenbacheri*	Larvae	92.4	G	N balance	[51]
	*Eucheira socialis*		93.5	G	N balance	[52]
	*Laniifera cyclades*	Larvae	98.9	G	N balance	[51]
	*Xyleutes redtembacheri*		92.4	G	N balance	[52]
Orth.	*Acheta domesticus*		91.7	G-I	N balance	[58]
	*Acheta domesticus*		85.3	G	N balance	[50]
	*Acheta domesticus*	Adult	57 *	G-I	N balance	[60]
	*Acheta domesticus*		65.5	G-I	N balance	[65]
	*Gryllus bimaculatus*	Adult	44 *	G-I	N balance	[60]
	*Locusta migratoria*	Adult	40 *	G-I	N balance	[60]
	*Ruspolia differens*	Adult	82.3	G-I	N balance	[59]
	*Sphenarium histrio*	Adult	85.6	G	N balance	[52]
Ref.	Fish meal		85.7	G-I	N balance	[58]
Feed	Fish meal		84.9	G	N balance	[46]
	Fishmeal: high protein		68 *	G-I	N balance	[60]
	Fishmeal: low protein		48 *	G-I	N balance	[60]
	Poultry meat meal		87.9	G-I	N balance	[58]
	Soyabean meal		94.7	G-I	N balance	[58]
	Soyabean meal		60.8	G-I	N balance	[65]
	Soyabean meal		61 *	G-I	N balance	[60]
Blat.	*Blaptica dubia*		32.5 *	O-G-I	DH (TNBS)	[56]
	*Gromphadorhina portentosa*		33.5 *	O-G-I	DH (TNBS)	[56]
Coleop.	*Alphitobius diaperinus*	Larvae	15.8 *	G-I	DH (pH-Stat)	[39]
	*Eulepida mashona*	Adult	30.6	O-G-I	DH (OPA)	[55]
	*Protaetia brevitarsis*	Larvae	54.9	G-I	DH (OPA)	[61]
	*Tenebrio molitor*	Larvae	14.9 *	G-I	DH (pH-Stat)	[39]
	*Tenebrio molitor*	Larvae	14.8	O-G-I	DH (TNBS)	[2]
	*Tenebrio molitor*	Larvae	54 *	G-I	DH (OPA)	[30]
	*Zophobas morio*	Larvae	28 *	O-G-I	DH (TNBS)	[56]
Dipt.	*Hermetia illucens*	Larvae	22 *	G-I	DH (pH-Stat)	[39]
Hemi.	*Carbula marginella*	Adult	81.7 *	I	DH (pH-Stat)	[53]
Lepid.	*Cirina butyrospermi*	Larvae	82.3 *	I	DH (pH-Stat)	[53]
Orth.	*Amphiacusta annulipes*		15.8	O-G-I	DH (TNBS)	[56]
	*Gryllodes sigillatus*		32 *	O-G-I	DH (TNBS)	[2]
	*Henicus whellani*	Adult	29.7	O-G-I	DH (OPA)	[55]
	*Locusta migratoria*		36.3	O-G-I	DH (TNBS)	[55]
	*Schistocerca gregaria*		30.5 *	O-G-I	DH (TNBS)	[2]
Ref.	Beef meal		50.6	G-I	DH (OPA)	[61]
Feed	Whey protein		34 *	O-G-I	DH (OPA)	[55]

* Aproximate values obtained from the figures. ICPD: in vitro crude protein digestibility. Blat: Blattodea. Coleop: Coleoptera. Dipt: Diptera. Hemi: Hemiptera. Hyme: Hymenoptera. Lepid: Lepidoptera. Orth: Orthoptera. DH: degree of hydrolysis. O: oral. G: gastric. I: intestinal. DH: degree of hydrolysis. N: nitrogen. aa: amino acids.

**Table 4 insects-13-00682-t004:** Effect of different processing methods on insect IVCPD.

Author	Specie/Ingredient	Order	Stage	Processing	Phases	% Digest.	ICPD Method
[61]	*Protaetia brevitarsis*	Coleop.	Larvae	Control	G-I	54.9	DH (OPA)
	*Protaetia brevitarsis*	Coleop.	Larvae	Defatted	G-I	46.6	DH (OPA)
	Beef loin				G-I	50.6	DH (OPA)
[3]	*Hermetia illucens*	Dipt.	Larvae	Conventional dried (60 °C drying oven)	O-G-I	82.0	aa balance
	*Hermetia illucens*	Dipt.	Larvae	Microwave dried (500 W 15’)	O-G-I	75.0	aa balance
	*Hermetia illucens*	Dipt.	Larvae	Conventional dried (60 °C drying oven)	O-G-I	90.0	aa balance
	*Hermetia illucens*	Dipt.	Larvae	Microwave dried (500 W 15’)	O-G-I	84.0	aa balance
[59]	*Macrotermes subhylanus*	Hymen.	Adult	Control	G-I	90.5	N balance
	*Macrotermes subhylanus*	Hymen.	Adult	Toast	G-I	90.4	N balance
	*Macrotermes subhylanus*	Hymen.	Adult	Toast + solar drying	G-I	90.1	N balance
	*Macrotermes subhylanus*	Hymen.	Adult	Solar drying	G-I	90.1	N balance
	*Ruspolia differens*	Orthop.	Adult	Control	G-I	82.3	N balance
	*Ruspolia differens*	Orthop.	Adult	Toast	G-I	80.1	N balance
	*Ruspolia differens*	Orthop.	Adult	Toast + solar drying	G-I	76.4	N balance
	*Ruspolia differens*	Orthop.	Adult	Solar drying	G-I	79.6	N balance
[31]	*Tenebrio molitor*	Coleop.	Larvae	Control (raw)	G-I	85	N balance
	*Tenebrio molitor*	Coleop.	Larvae	Vacuum-cooked	G-I	90.5	N balance
	*Tenebrio molitor*	Coleop.	Larvae	Fried	G-I	87.2	N balance
	*Tenebrio molitor*	Coleop.	Larvae	Boiled	G-I	90.1	N balance
	*Tenebrio molitor*	Coleop.	Larvae	15’ oven-cooked	G-I	91.5	N balance
	*Tenebrio molitor*	Coleop.	Larvae	30’ oven-cooked	G-I	90.4	N balance
[65]	*Acheta domestica*	Orthop.	Adult	Raw	G-I	65.5	N balance
	*Acheta domestica*	Orthop.	Adult	Oven at 150 °C	G-I	59.3	N balance
	*Acheta domestica*	Orthop.	Adult	Oven at 200 °C	G-I	61.1	N balance
	*Acheta domestica*	Orthop.	Adult	Autoclaved	G-I	59.5	N balance
	*Tenebrio molitor*	Coleop.	Larvae	Raw	G-I	72.5	N balance
	*Tenebrio molitor*	Coleop.	Larvae	Oven at 150 °C	G-I	64.1	N balance
	*Tenebrio molitor*	Coleop.	Larvae	Oven at 200 °C	G-I	63.9	N balance
	*Tenebrio molitor*	Coleop.	Larvae	Autoclaved	G-I	59.5	N balance
	Soybean			Raw	G-I	60.8	N balance
	Soybean			Vapor cooked	G-I	76.3	N balance
[62]	*Tenebrio molitor*	Coleop.	Larvae	Control	G-I	60 *	N balance
	*Tenebrio molitor*	Coleop.	Larvae	Oven cooking (70 °C for 30’)	G-I	75 *	N balance
	*Tenebrio molitor*	Coleop.	Larvae	Deep fry	G-I	40 *	N balance
[49]	*Hermetia illucens*	Dipt.	Prepupae	Blanched	G	32	DH (OPA)
	*Hermetia illucens*	Dipt.	Prepupae	Frozen	G	16.5	DH (OPA)
	*Hermetia illucens*	Dipt.	Prepupae	Frozen and Blanched	G	18	DH (OPA)
[2]	*Tenebrio molitor*	Coleop.	Larvae	Raw	O-G-I	14.8	DH (TNBS)
	*Tenebrio molitor*	Coleop.	Larvae	Boiled	O-G-I	31.4	DH (TNBS)
	*Tenebrio molitor*	Coleop.	Larvae	Baked	O-G-I	11.3	DH (TNBS)
	*Gryllodes sigillatus*	Orthop.	Adult	Raw	O-G-I	32 *	DH (TNBS)
	*Gryllodes sigillatus*	Orthop.	Adult	Boiled	O-G-I	26.5 *	DH (TNBS)
	*Gryllodes sigillatus*	Orthop.	Adult	Baked	O-G-I	37.8	DH (TNBS)
	*Schistocerca gregaria*	Orthop.	Adult	Raw	O-G-I	30.5 *	DH (TNBS)
	*Schistocerca gregaria*	Orthop.	Adult	Boiled	O-G-I	37.7	DH (TNBS)
	*Schistocerca gregaria*	Orthop.	Adult	Baked	O-G-I	32 *	DH (TNBS)
[39]	*Tenebrio molitor*	Coleop.	Larvae	Raw	G-I	14.9 *	DH (pH-Stat)
	*Tenebrio molitor*	Coleop.	Larvae	Sulfite	G-I	18 *	DH (pH-Stat)
	*Tenebrio molitor*	Coleop.	Larvae	Blanched	G-I	12.5 *	DH (pH-Stat)
	*Alphitobius diaperinus*	Coleop.	Larvae	Raw	G-I	15.8 *	DH (pH-Stat)
	*Alphitobius diaperinus*	Coleop.	Larvae	Sulfite	G-I	13 *	DH (pH-Stat)
	*Alphitobius diaperinus*	Coleop.	Larvae	Blanched	G-I	15.1 *	DH (pH-Stat)
	*Hermetia illucens*	Dipt.	Larvae	Raw	G-I	22 *	DH (pH-Stat)
	*Hermetia illucens*	Dipt.	Larvae	Sulfite	G-I	26 *	DH (pH-Stat)
	*Hermetia illucens*	Dipt.	Larvae	Blanched	G-I	16 *	DH (pH-Stat)
[55]	*Eulepida mashona*	Coleop.	Adult	-	O-G-I	30.6	DH (OPA)
	*Eulepida mashona*	Coleop.	Adult	Boiled 30’	O-G-I	27.5 *	DH (OPA)
	*Eulepida mashona*	Coleop.	Adult	Boiled 60’	O-G-I	25 *	DH (OPA)
	*Eulepida mashona*	Coleop.	Adult	Boiled 30’ + Roasted	O-G-I	25.5 *	DH (OPA)
	*Eulepida mashona*	Coleop.	Adult	Roasted	O-G-I	30.4 *	DH (OPA)
	*Henicus whellani*	Orthop.	Adult	-	O-G-I	29.7	DH (OPA)
	*Henicus whellani*	Orthop.	Adult	Boiled 30’	O-G-I	24.2	DH (OPA)
	*Henicus whellani*	Orthop.	Adult	Boiled 60’	O-G-I	25 *	DH (OPA)
	*Henicus whellani*	Orthop.	Adult	Boiled 30’ + Roasted	O-G-I	21.5 *	DH (OPA)
	*Henicus whellani*	Orthop.	Adult	Roasted	O-G-I	24.7	DH (OPA)
	whey protein				O-G-I	34 *	DH (OPA)
[64]	*Hermetia illucens*	Dipt.	Larvae	Larvae + Wheat non extruded	G-I	93.0	N balance
	*Hermetia illucens*	Dipt.	Larvae	Larvae + Wheat extruded 60 °C	G-I	94.0	N balance
	*Hermetia illucens*	Dipt.	Larvae	Larvae + Wheat extruded 70 °C	G-I	94.0	N balance
	*Hermetia illucens*	Dipt.	Larvae	Larvae + Wheat extruded 80 °C	G-I	94.0	N balance
	*Hermetia illucens*	Dipt.	Larvae	Larvae + Wheat extruded 90 °C	G-I	94.0	N balance
[48]	Control			Aquafeed extruded (shrimp meal 100%) (20% moisture)	G	57 *	N balance
	*Acheta domestica* + feed	Orthop.	Adult	Aquafeed extruded (AcD 25% + shrimp meal 75%) (20% moisture)	G	72 *	N balance
	*Acheta domestica* + feed	Orthop.	Adult	Aquafeed extruded (AcD 50% + shrimp meal 50%) (20% moisture)	G	69 *	N balance
	*Acheta domestica* + feed	Orthop.	Adult	Aquafeed extruded (AcD 75% + shrimp meal 25%) (20% moisture)	G	65 *	N balance
	*Hermetia illucens* + feed	Dipt.	Prepupae	Aquafeed extruded (HI 25% + shrimp meal 75%) (20% moisture)	G	59 *	N balance
	*Hermetia illucens* + feed	Dipt.	Prepupae	Aquafeed extruded (HI 50% + shrimp meal 50%) (20% moisture)	G	53 *	N balance
	*Hermetia illucens* + feed	Dipt.	Prepupae	Aquafeed extruded (HI 75% + shrimp meal 25%) (20% moisture)	G	56 *	N balance
	Control			Aquafeed extruded (shrimp meal 100%) (30% moisture)	G	54 *	N balance
	*Acheta domestica* + feed	Orthop.	Adult	Aquafeed extruded (AcD 25% + shrimp meal 75%) (30% moisture)	G	58 *	N balance
	*Acheta domestica* + feed	Orthop.	Adult	Aquafeed extruded (AcD 50% + shrimp meal 50%) (30% moisture)	G	60 *	N balance
	*Acheta domestica* + feed	Orthop.	Adult	Aquafeed extruded (AcD 75% + shrimp meal 25%) (30% moisture)	G	57 *	N balance
	*Hermetia illucens* + feed	Dipt.	Prepupae	Aquafeed extruded (HI 25% + shrimp meal 75%) (30% moisture)	G	63 *	N balance
	*Hermetia illucens* + feed	Dipt.	Prepupae	Aquafeed extruded (HI 50% + shrimp meal 50%) (30% moisture)	G	56 *	N balance
	*Hermetia illucens* + feed	Dipt.	Prepupae	Aquafeed extruded (HI 75% + shrimp meal 25%) (30% moisture)	G	59 *	N balance
[54]	*Tenebrio molitor*	Coleop.	Larvae	Larvae 0% + Wheat 100% extruded	G-I	91	DH (OPA)
	*Tenebrio molitor*	Coleop.	Larvae	Larvae 10% + Wheat 90% extruded	G-I	91	DH (OPA)
	*Tenebrio molitor*	Coleop.	Larvae	Larvae 20% + Wheat 80% extruded	G-I	89	DH (OPA)
[47]	*Tenebrio molitor*	Coleop.	Larvae	Larvae 0% + feed 100% extruded 150 °C 50% moistture	G	76.6	N balance
	*Tenebrio molitor*	Coleop.	Larvae	Larvae 30% + feed 70% extruded 140 °C 40% moistture	G	98.8	N balance

ICPD: in vitro crude protein digestibility. O: oral. G: gastric. I: intestinal. DH: degree of hydrolysis. DM: dry matter. N: nitrogen. aa: amino acids. * Approximate values obtained from the figures.

## Data Availability

No new data were created or analyzed in this study.

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
