# Peer review of "In Vitro Crude Protein Digestibility of Insects: A Review"

_insects, 2022, doi:10.3390/insects13080682_

Round 1

Reviewer 1 Report

The review is, in general, interesting. Some minor changes are required.

L. 27-27 "which method offer..." please rewrite this period. Something is wrong.

L. 42: "do to this..." This? What?

L. 56 -57. In the cited studies the values are referred not only to whole insects but also to meal. Probably some defatted meals. Among the factors affecting the percentage of protein, the de-fatting must be considered.

L. 90. Add a reference.

Author Response

Thank you.

Reviewer 2 Report

The article entitled: “In Vitro Crude Protein Digestibility of Insects: a Review” is a complete work. The authors correctly report how the two databases used for the paper were obtained. A very interesting aspect of the paper is the two-way approach that the authors used: firstly they investigate the crude protein digestibility of different insects species, then they deeply investigate the effects that the processing processes have on insect protein digestibility.

General comments:

In the article there are some typos, for this reason few English corrections are needed.

Since the Hermetia illucens and the Tenebrio molitor are the two most cites insect species in the paper, I suggest to use their acronyms (HI and TM, or BSF and YM). Moreover, pay attention to the scientific names that must be written in italics.

Finally, please check and correct all the tables and the data reported in the paper since number are sometimes reported with the comma instead of the dot.

Specific comments:

Line 17: Please, change discusion with discussion

Line 17: “in vitro” should be written in italics

Line 19: This phrase is not very clear, please specify what do you mean with “comparisons”

Lines 20-21: From this phrase it seams that the subject is the protein content, while it is insects. Please correct this sentence.

Line 26: Digestibility is not only affected by the processing of the raw materials but also depends on the animal species considered. Please, pay attention on this aspect through the paper.

Line 49-51: You can also add the new Commission Regulation (EU) 2021/1372 of 17 August 2021 authorizing insects also for poultry and swine

Line 52: here there is a typo: black soldier

Line 58: please change to “variability could be due to…”

Line 59: crude protein instead of Crude Protein

Table 1: Specify the acronym for crude protein in the table description

Line 87: This is not really true. There are digestibility studies specifically created to test the ingredient digestibility in animals (poultry for example). I suggest to remove this phrase or to rephrase it.

Line 119: Use “species” instead of “type”

Lines 159-162: Since the aim of the paper is the evaluation of crude protein digestibility, I think that this information is not necessary and can confuse the reader. I suggest to remove this paragraph.

Line 231: please, indicate the ADF extended

Line 237: Correct the citation n. 71

Line 266 and 267: please correct “y” with “and”

Lines 271, 275, 284, 289: It should be better to write “Order” extended

Lines 284-288: Please rephrase these sentences. It is not clear which “other species” refers to.

Lines 291-292: For these data I think that it should be better to use a mean ± standard dev.

Line 293: Since H. illucens is one of the most promising insect species for animal feeding, I suggest to include a paragraph about Diptera

Line 297: BAPs ?

Table 4: I understand that the insects’ species are organized by author. However, it should be better to indicate the author in the first column in order to make the table clearer.

Line 333: Please specify what do you mean for “control”

Line 342: here there is a typo: “… meal of…”

Line 348: for 60°C for how many minutes?

Lines 372-373: do you have information about the timing?

Line 406: please correct reference n. 92

Lines 433 and 434: Since it is “digestibility of organic matter” I suggest to use the acronym “DOM” instead of “DMO”

Line 451: please correct reference n. 55

Author Response

Thank you.
